

# Synthetic Weather Diaries: Concept and Application to Swiss Weather in 1816

Stefan Brönnimann[1,2]

[1] Institute of Geography, University of Bern, 3012 Bern, Switzerland

[2] Oeschger Centre for Climate Change Research, University of Bern, 3012 Bern, Switzerland

*Correspondence to*: S. Brönnimann (stefan.broenniman@giub.unibe.ch)

**Abstract.** Climate science is about to produce numerical daily weather reconstructions based on meteorological measurements for Central Europe 250 years back. Using a pilot reconstruction covering Switzerland at 2x2 km2 resolution for 1816, this paper presents methods to translate numerical reconstructions and derived indices into text describing daily weather and the state of vegetation. This facilitates comparison with historical sources and analyses of effects of weather on different aspects of life. The translation, termed "synthetic weather diary" could possibly be used to train machine learning approaches to do the reverse: reconstruct past weather from categorized text entries in diaries.

## 1 Introduction

The past decade has seen tremendous advances in numerical weather reconstructions. They were enabled by new numerical techniques such as data assimilation (Compo et al. 2011), combined with a new demand arising from new scientific questions (better understanding variability), new societal needs (prepare for extreme weather events in the future), and the more widespread use of numerical modelling in climate impact research. In this situation, historical instrumental data – but also documentary weather data - become once again valuable for science (Allan et al., 2011). Based on digitised historical instrumental data, model chains can be built (e.g., numerically simulating the damage of past storm or flood events, see Stucki et al., 2015, 2018) and analysed together with historical sources (e.g., Allan et al., 2016; Veale et al., 2017).

In addition to data assimilation, providing global weather data at coarse resolution back to the early 19th century (Slivinski et al. 2019), also other techniques such as analog resampling of regional weather fields (Caillouet et al., 2019; Devers et al., 2020; Pfister et al., 2020) have been used to reconstruct local daily weather 150-200 years back in time, with the potential to go even further back. These reconstructions provide a resource not just for climate science, but also for historians. Depending on the application, a translation between numerical weather data and the descriptive text format in which historical observations and weather diaries are typically written would be beneficial. In this paper I present a first step in this direction, termed "synthetic weather diary".

Turned into categorized text, numerical weather reconstructions could supplement historical sources with weather descriptions, much like weather reports today. They could provide, for instance, information on the day-to-day weather for a specific journey or during a military operation. In addition to the individual measurements, useful information for historians could be gained from specific indices based on these daily data. Such indices could provide information on the freezing of water bodies, the state of vegetation, or drought conditions.

The translation makes numerical reconstructions and observations directly comparable, which is not only useful for historians, but also for climate sciences. For instance, generating synthetic weather diaries from numerical data



in recent decades could be used to train machine learning algorithms to provide the weather pattern (e.g., the
weather type or even a full spatial field). A trained algorithm could then be used to classify daily weather in the
past based on categorized information from historical weather diaries.
Codification of descriptive weather information is a core work of historians of climate (Riemann et al., 2015). A
next step then is to establish an ordinal scale, which has been attempted mostly at the monthly or seasonal scale.
So-called "Pfister indices" (Pfister et al., 2018) are often used and categorize weather in three-point (-1, 0, 1) or
seven-point (-3, -2, -1, 0, 1, 2, 3) indices, with corresponding designations such as "extremely cold", "cold", etc.
(Pfister, 1999). Calibrating such indices with measurement-based time series in order to calibrate climate
reconstructions implies a similar translation from numerical data to text, though on the monthly or seasonal scale.
In this paper I describe a pilot study to generate synthetic weather diaries based on daily weather reconstructions
for Switzerland for the year 1816, known as a „Year Without a Summer". Weather and climate during this year
were affected by the eruption of Tambora in Indonesia in 1815 (Raible et al., 2016; Brönnimann and Krämer,
2016). The summer was very cold and rainy, particularly in Central Europe (Auchmann et al., 2012; Luterbacher
and Pfister, 2015; Veale and Endfield, 2016), although the adverse weather conditions were only partly due to
direct effects of the eruption. The "Year Without a Summer" of 1816 is arguably among the most prominent
climate events of the past 350 years. Concurring with increased vulnerability after the Napoleonic wars (due to
political instability, economic shifts after the end of the continental system, high unemployment rates, and
inadequate governance), the societal consequences of this weather event were severe, particularly in Switzerland
(Krämer, 2015; Behringer, 2016). Further weather-related effects were related to snow accumulation, such as
avalanches in the following winter (Rohr, 2015) and a flood event in early summer 1817, although the snowmelt
contribution was large only close to the Alps (Rössler and Brönnimann, 2018). The Tambora eruption was one of
at least five large eruptions within a relatively short period, which together had profound effects on the global
climate system, including monsoons, and on Alpine glaciers (Brönnimann et al., 2019).
In the following I describe the weather reconstruction, the translation into text as well as the generation of impact-
related indices. Then I evaluate the synthetic diaries by comparison with daily observations (which are already in
categorized form) and monthly weather summaries as well as brief daily weather notes written down on Lord
Byron's famous journey through Switzerland (Byron, 1839) in September 1816. A discussion then follows on the
use of "synthetic weather diaries" in history and science. The paper ends with brief conclusions.
**2 Data**
**2.1 Numerical reconstructions**
The daily weather reconstructions for 1816 used in this paper are taken from Flückiger et al. (2017) and cover
Switzerland. They were produced with the specific aim of numerically modelling potential harvest yields during
the "Year Without a Summer" of 1816 in Switzerland. Furthermore, they were subsequently used to numerically
model the flood events of 1817 (Rössler and Brönnimann, 2018). In the following, I give a brief summary of the
reconstruction technique. Note that the reconstruction was not designed for day-to-day accuracy. Nevertheless, it
best serves the purpose of presenting the concept of synthetic weather diaries as it is the earliest, high-resolution
spatial weather reconstruction available at the moment of writing and also because the "Year Without a Summer"
of 1816 is well documented and of particular interest.



The reconstruction is based on an analog resampling. The analog pool consists of 59 years of daily 2 x 2 km$^2$
resolved fields of minimum and maximum temperature ($T_{min}$ and $T_{max}$, which we averaged to daily mean
temperature, $T_{mean}$, for this study) and precipitation from the data sets TminD, TmaxD, and RhiresD (Frei, 2014),
reaching back to 1961. Solar irradiance fields were produced using a k-nearest neighbour interpolation. From this
pool, the day that is closest to a historical target day, according to all measurements available at that historical day,
was chosen, as is detailed in the following. The process was repeated for each day, and the sequence of closest
analogs constitutes the reconstruction.
Three stations were used to define a measure of distance between analog pool and historical target day: Geneva,
Delémont (both in Switzerland), and Hohenpeissenberg (Bavaria). These were the only three stations within or in
the vicinity of Switzerland with daily or subdaily data at the time the reconstruction was produced. We used $T_{min}$
and $T_{max}$ for Geneva and Hohenpeissenberg, $T_{mean}$ for Delémont and precipitation for Geneva. The Eucledian
distance was chosen as the distance measure. We accounted for a change in temperature between the historical
period and the pool-of-analog period, as detailed in Flückiger et al. (2017).
The analog selection then proceeded in several steps: The chosen analog must be in the same season as the target
day (calendar day ±30 days) and must be of the same weather type (according to Auchmann et al., 2012). Then,
the closest day in the analog pool according to Eucledian distance between the station measurements of the
historical target day and every day in the pool of analogs was chosen. Finally, the temperature change was
subtracted from the analog temperature to obtain the reconstruction for the past (precipitation and irradiance were
not changed).
Note that a much improved reconstruction (relying on much more data and using a post-processing step that
additionally corrects the best analog towards the observations) is available from 1864 onward (Pfister et al., 2020).
It is used in this paper only briefly to assess the effect of the reconstruction quality on the agreement between the
synthetic weather diary and independent weather notes.
In addition to the reconstructed fields, I also use Swiss weather types, which are available for every day back to
1763 (Schwander et al. 2017). The data set provides for each day the most likely of seven weather types, along
with its probability. Each weather type can be equated to a short meteorological description of the weather. The
seven weather types (termed CAP7, where CAP stands for Cluster Analysis of Principal Components) are based
on the so-called CAP9 weather types of MeteoSwiss (Weusthoff, 2011) which encompass nine types. The
reduction to seven types (by combining existing types) was necessary as in the early decades, some types could
not be well discriminated, Table 1 shows an overview.
**2.2. Data for comparison: Non-instrumental observations**
The gridded reconstruction is based only on instrumental measurements from three stations. In a recent project we
have uncovered (Pfister et al., 2019) and digitized (Brugnara et al., 2020) many more instrumental series for
Switzerland back to the early 18th century, such that we now have ten series for 1816. This effort was part of a
global effort to uncover more historical instrumental data (Brönnimann et al., 2019b), a lot of which is currently
being digitized. With these data, more accurate reconstructions will be produced in the future. The Swiss data
could be used to independently evaluate the reconstruction used here, but I touch on this only briefly as it is not
the goal of this paper. Rather, most of these additional series also have categorized daily weather observations,
and some have descriptive monthly summaries. In this paper I focus on these entries for three series: Geneva,





Aarau, and St. Gall. These three series are described in the following; they can be downloaded from EURO-
CLIMHIST (Pfister et al., 2017).
The Aarau weather data, published in the journal *Archiv der Medizin Chirurgie und Pharmazie* (Zschokke, 1817)
contain twice or three times daily instrumental (pressure, temperature) and non-instrumental (precipitation, sky
cover) information (Faden et al., 2020). An example for September 1816, is given in Figure 1. The weather data
were recorded by Heinrich Zschokke (1771-1848), a German-born teacher and politician. The record was
continued by his son and overall covers almost 60 years.
Likewise, the St. Gall record contains instrumental (temperature, pressure) and non-instrumental (precipitation,
sky cover) information, recorded twice daily. The observations were made by pharmacist Daniel Meyer (1778-
1864) from 1812 to 1832 and continued by other, unknown observers (Hürzeler et al., 2020). The data were
published in the journal *Der Erzähler* (Meyer, 1816, 1817).
The series from Geneva (Auchmann et al., 2012) was observed by Marc-Auguste Pictet (1752-1825), a scientist
and publisher from Geneva. He was director of the Geneva Observatory; in fact, he may rightly be called a
meteorologist. His observations, published monthly in the journal "Bibliothèque universelle" (Pictet, 1816), also
include monthly summaries with notes on the state of vegetation. Note that the instrumental measurements from
Geneva were used to produce the daily numerical reconstruction, so they are not independent. Therefore, I do not
compare the daily entries here, but rather the monthly summaries.
Finally, I also briefly use weather observations made by a person named Furrer in Winterthur from 1849-1867
(Pfister et al., 2019). The data were taken from the Zurich State Archive (Furrer, n. y.). They are used to test my
approach in a later period, when better reconstructions are available.

**2.3 Data for comparison: Weather diary**

Synthetic weather diaries can also be generated for journeys. To test this, I used the travel diary of Lord Byron
during his famous voyage from Lake Geneva through the Bernese Oberland in September 1816. I extracted several
weather related statements from his travel journal (Byron, 1839) and extracted the same information, in space and
time, in the numerical reconstruction.

**3 Method**

**3.1 General concept and reference period**

In order to make the weather reconstructions as useful as possible to non-climatologists, while allowing to better
compare them to weather observation notes, I structured the data in a similar way as found in many observation
books. Daily values and descriptors are listed, and for each month a summary is given:
• For each day, the absolute number is given for $T_{mean}$ and precipitation, accompanied by additional
information (relating to a reference period, see below) and a set of descriptive qualifiers concerning each
variable and the weather type.
• For each month, monthly statistics are given for $T_{mean}$ and precipitation in numerical form. Additionally,
monthly indices are calculated and given numerically. Again, this section is accompanied by descriptive
qualifiers for both, the monthly weather and the indices.
Observers might sometimes report on temperature in an "absolute" manner (e.g., referring to freezing), but often
also in a relative way (e.g., "very cold day") based on their own experience and perhaps in some cases alluding to





societal memory. This requires information on the observers and observation context. For this study this implies
putting absolute numbers in the context of a reference period. For consistency, I use the same reference period as
in Flückiger et al. (2017): the period 1800-1820 without the volcanically perturbed years 1809-1911 and 1815-
1817. Fields of reference period mean temperature for every calendar day are available from Flückiger et al.
(2017), so results presented here can be compared with that study. However, obviously a historical observer would
not have a memory of the future.
Note that Flückiger et al. (2017) did not reconstruct every day in the historical reference period. Rather, they
defined a present-day reference period, 1982-2009, from which they subtracted the difference (in terms of a
seasonal cycle) between the periods 1800-1820 and 1982-2009 based on instrumental observations from the three
mentioned stations for the historical reference period. For precipitation and solar irradiance, no change was added
and thus 1982-2009 is considered as a reference period for these two variables. In this paper, for consistency, I
follow the same approach.
Historical observers, in their reporting, might account for the changing variability in the course of the seasons. For
instance, variability is larger in winter than in summer. Temperature on a "cold" summer day might be less below
average (in degrees Celsius) than on a "cold" winter day. I therefore standardized the anomalies, again using the
1982-2009 standard deviation (calculated for each calendar day and then smoothed by fitting the first two
harmonics of the seasonal cycle). Likewise, monthly averages or monthly statistics were expressed as standardized
anomalies by using the reference period annual cycle and standard deviations calculated per calendar month.
**3.2 Obtaining daily weather descriptions**
The first step to obtain daily weather descriptions is to establish a taxonomy that eventually allows a comparison
between observations and numerical reconstructions. The target taxonomy must be reducible to the observed
taxonomy, but ideally contains additional information. The observations by both Zschokke and Meyer were
already extremely standardised. With respect to precipitation, the main categories are rain, snow, or an empty field
(standing for dry). In the case of Zschokke, we also (rarely) find the terms "Schneeregen" (mix of snow and rain)
and "Staubregen" (most likely: drizzle[1]). In the case of Winterthur, which was used for testing the method in 1865,
we also find "Nebelregen", "Nebel" and "neblig" (fog rain, fog, foggy; for the comparison we assume that
precipitation amounts are below the detection threshold chosen in the next Section.).
In short, Zschokke and Mayer both provide basically three categories (rain, snow, or dry), two or three times per
day. The synthetic weather diary has only daily resolution (so the observations need to be aggregated for
comparison), but information can be categorized into more classes which can then be aggregated. The definitions
of the classes are indicated in Table 2 and described in more detail in the following.
For all standardized anomalies (daily or monthly) we use a seven-point scale defined in Table 3. Note that this
scale deviates from similar seven-point scales as defined, e.g., by Pfister et al. (2018) which is also included in
Table 3. This is because on the daily scale, a non-linear (in terms of the underlying variable; the scale is almost
linear in terms of probabilities) categorization as implied by the "Pfister indices" seems hard to achieve; it would
require detecting rather subtle changes close to the average. Therefore a linear scale is preferred. The basic
categories -*x* or *x* (e.g., "cold" or "warm") are similar in the two classifications, with thresholds roughly near the

---

[1] „Staubregen" is described as drizzle in many contemporary dictionaries (e.g., Adelung 1811). However, the term was also used for dust fall (e.g., Saharan dust events). For the comparison we assume that the amount in any case would be below the detection threshold that is later chosen.



quartiles, but there is a large discrepancy in the use of the term "extreme". There is approximate agreement of my
scale with the likelihood scale in the IPCC calibrated language (IPCC, 2013) where "likely" and "very likely"
refer to 66% and 90% cumulative probability (in my scale *"x"* and "very *x*" refer to 69.1% and 93.3%,
respectively). However, the scale can easily be adapted, and indeed should be adapted, for other applications.
For the daily values, the following information is given:

- For $T_{mean}$ the synthetic weather diary contains the absolute values, the anomaly from a contemporary
reference period, and the standardized anomaly. The taxonomy is based on the latter, using the seven-
point scale (Table 3): values < -2.5 are termed "extremely cold", -2.5 to -1.5 "very cold", -1.5 to -0.5
"cold", -0.5 to 0.5 "average", 0.5 to 1.5 "warm", 1.5 to 2.5 "very warm", and >2.5 "extremely warm".

- For precipitation the synthetic diary gives the absolute value as well as the qualifier "dry" (<1 mm),
"slight rain" (1 to 5 mm), "rain" (5 to 15 mm) and "heavy rain" (>15 mm). The thresholds are chosen
arbitrarily. For the case of Geneva, 69% of the days in the reference period are "dry". Of the remaining
days, half have "slight rain", 36% have "rain" and 14% "heavy rain". The sensitivity to the choice of the
thresholds is analysed later. If daily mean temperature was below 2 °C (following Zubler et al., 2014) the
qualifiers "slight snowfall", "snowfall", "heavy snowfall" are used instead.

- Sky conditions are given as text. For this, irradiance was first expressed as fraction of the maximum
possible value for the corresponding calendar day. The latter was approximated here by the simple
function $100 * (1.4 \ W/m^2 \cdot cos(w))$, where $w$ is the angle corresponding to the calendar day centred around
the solstices. If the fraction is higher than 0.66 and no precipitation is reconstructed, the sky is descried
as "clear"; if it is below 0.66, or if there is precipitation, "partly cloudy", if it is below 0.33, "cloudy".

- Finally, I provide the most likely weather type for that day from the Swiss CAP7 weather statistics
(Schwander et al., 2017), along with the probability of that weather type on that day and a text description
of the type. Note that the weather type cannot directly be compared with the (often observed) wind
direction unless the local situation is very well understood. However, for future applications, wind would
be an important component of a synthetic weather diary.
Note that irradiance was calculated from an interpolation of few station data and should only be analysed closed
to current weather stations, which is the case for the four extracted weather diaries. However, spatial field cannot
be analysed, and I do not provide numerical values in the synthetic weather diaries.

**3.3 Monthly weather summaries**

For each month, the following information is given:

- Mean $T_{mean}$ of the month, again along with the deviation from the reference and a qualifier (7-point scale,
Table 3), number of freezing days ($T_{mean}$ < 0 °C)

- Precipitation sum (also expressed as percentage of the reference for that calendar month) and number of
rain or snow days,

- Monthly counts of the 7 weather types (also expressed as percentage of the corresponding relative
frequency for that calendar month in the reference period), and

- Three derived indices described below: Growing Degree Days, maximum 10-day value of Freezing
Degree Days, and water balance, each given as absolute value and qualifiers
This information can further be condensed, if necessary. The following monthly indices are used:





•   Growing Degree Days (GDD): The cumulative sum of $T_{mean}$ above 4 °C (starting on 1 January) is a
measure for suitability for crop growth. Each month I give the value for that month (with reference) as
well as the delay at the end of the month with respect to the median in the reference period, and a
qualitative description ("extremely early" to "extremely late", according to standardised anomalies using
the 7-pt scale). While leaf unfolding or flowering quite generally depends on GDD, crop maturity (and
thus harvest dates) or leaf colouring are more species dependent and depend on other factors. Therefore,
GDD delays and qualifiers are only given for the months of March to July.

•   Freezing Degree Days (FDD): The cumulative sum of $T_{mean}$ below 0 °C (with a positive sign),
accumulated over the 10 previous days. For the months January to April and October to December, the
maximum of this 10-day index per month ($max_{10}FDD$) is given. If maximum FDD exceeds 50 °C, then a
note "small lakes frozen" is indicated, if it exceeds 90 °C, "large lakes frozen" is indicated. These
thresholds are in approximate agreement with Franssen and Scherrer (2008), but, again, would have to be
adapted for each application.

•   A monthly index of the water balance (precipitation minus potential evapotranspiration, P-E) is calculated
by making use of the precipitation amount and $T_{mean}$, from which potential evapotranspiration is
calculated using the Thorntwaite (1948) formula. The monthly balance is standardized and then described
in each month with the 7-pt scale ("extremely wet" to "extremely dry").

**4 Results**
**4.1 Non-instrumental observations for Aarau and St. Gall**
The synthetic weather diaries for Aarau and St. Gall are given in the electronic supplement. Their performance in
terms of temperature can be measured by comparing the numerical values with the instrumental temperature series
from the two stations, which were not used in the reconstruction process and are thus independent. After
subtracting the mean annual cycle in the reference period from both series (by fitting the first two harmonics of
the seasonal cycle), I find a correlation of 0.81 and 0.72, respectively. Note that the measurements themselves
have errors. In view of that, the correlations indicate that even though the reconstruction is only based on three
stations, the temperature fields are quite reliable.
For rainfall and sky cover, I compare our synthetic diaries with the actual observations from the two stations. As
an example, the Aarau observations for July 1816 are shown in Fig. 1, the corresponding synthetic diary in Table
4. Both the observations and the synthetic diary indicate a particularly rainy month, but on a day-to-day scale there
are also clear differences. In the observation, there are only three days without any precipitation (20, 21 and 23
July), of which two are also dry in the synthetic diary. The latter gives eight "dry" days (of which four have zero
precipitation, four have less than 1 mm).
A plot comparing observations and synthetic weather diary for both stations for both precipitation and sky
conditions is shown in Fig. 2 (middle and bottom). While the agreement at the level of seasonal characteristics is
quite favourable – both synthetic weather diaries and observations confirm the high number of rainy days in
summer and also agree on less rainy periods – there are also important differences. For instance, the first half of
September is rather dry in the synthetic diaries (both Aarau and St. Gall), but many rainy days are reported at both
stations.

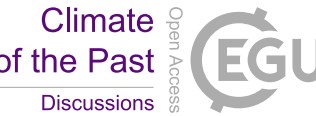

The agreement can be quantified with Spearman correlations by coding rain/no rain in the observations as 0 and 1
("Nebelregen", "Nebel", "neblig" and "Staubregen" were set to 0) and in the synthetic diary as 0 to 3 for "dry",
"slight rain/snowfall", "rain/snowfall", and "heavy rain/snowfall"). In this way I find correlations of around 0.25
for Aarau and St. Gall. Coding snow with a negative sign in both sources, correlations increase to slightly above
0.4 at both sites. Although highly significant, this agreement may not be good enough yet to be useful on a day-
to-day scale.
There are several causes for discrepancies: Errors in the reconstruction, errors in the observations, lack of
representativity of observations for a grid cell, and inadequate translation. The error in the reconstructions can be
partly assessed by comparison with similar analyses in a period after 1864, when better reconstructions are
available (Pfister et al., 2020). In these reconstructions, the error of precipitation was assessed by subsampling in
the 1961-2010 period. Correlations of 0.75-0.90 were found for most parts of Switzerland, which however
constitutes an upper-limit estimate as this analysis has no representativity error and a good quality of the underlying
measurements. A more realistic case is to analyse these reconstructions in the very early years of the Swiss
Meteorological network. Non-instrumental observations are available, for instance, for Winterthur. Comparing
rainfall in the first three years (1864-1866) in the same way (coding snow with a negative sign), I find a correlation
of 0.5. This analysis accounts for the same errors as in the case of 1816, the only difference being better
reconstructions. This estimate is an upper-limit of the quality that can possibly be reached 250 years back using
additionally digitised data (Brugnara et al., 2020). For the year 1865, the comparison is also shown in Fig. 2.
Clearly the agreement is better in this case. Most importantly, the difference between this relatively dry year and
the wet year 1816 is extremely clear.
The error of the translation could be assessed by changing the chosen thresholds. In the case of Aarau, the threshold
for precipitation could have been set too high in the synthetic diary. I tested all combinations of threshold of 0.5
or 1 mm, 5 or 8 mm, and 15 or 20 mm for the separation between the four classes for both Aarau and St. Gallen.
In fact, most other combinations gave slightly better results (*e.g.,* 0.5/8/20 mm), but differences were small. In any
case, for other applications the thresholds would have to be reconsidered.
Finally, the agreement between observed and synthetic sky conditions is very low. Correlations, defined similarly
as above, yield coefficients of 0.06 and 0.12 for St. Gall and Aarau, respectively, which is too low to be useful.
Visually, large differences become apparent between the observations at Aarau and St. Gall. Specifically, the
category "cloudy" ("bewölkt" in Zschokke, "trüb" or "neb. trüb" in Mayer) differs a lot between the sites at the
expense of "partly cloudy". More work and more care is required to obtain a good classification.
The weather type information can give additional information. For instance, every day but one (27 July) in the
example of July 1816 is attributed to a cyclonic weather type. This agrees well with the rainy character of the
month. Note also the frequent westerly winds noted in the observations, which is in accordance with westerly or
west-southwesterly weather types, although in that case knowledge of the local wind situation and the channelling
of winds is required.
**4.2 Monthly summaries for Geneva**
For testing the monthly summaries in the synthetic weather diaries, I compare them with the observations from
Geneva. Marc-Auguste Pictet, in his observations published in the "Bibliothèque universelle" also gives a monthly
summary. This sort of information is typical in historical weather sources. Here I compare the entries for the
months of March to September, which are most relevant for crops, in a qualitative way (Table 5). Note that for



this Table, for brevity's sake, the synthetic monthly summaries have been further simplified (e.g., not all weather
types are indicated, but only those that were anomalously frequent or infrequent).
The comparison (highlighted in italics) shows a relatively good agreement. Almost in all months, Pictet points to
the delay of vegetation, which is also seen in the synthetic diary based on growing degree days. The calculated
delay reaches 22 days in July (relative to the historical reference period). This is less than indicated by Pictet (one
month), which however refers to one comparison year. Agreement is also found with respect to most mentions of
temperature and rainfall. For instance, the reported "harsh temperatures" in March correspond to a cold month in
the synthetic diary, the "cold and rainy weather" in July compares well with the characterisation "very cold" and
"extremely wet" in the synthetic diary. Worse agreement is found for October, which according to Pictet was of
"remarkable beauty", but in the synthetic weather diary is characterised as "cold", though with below normal
rainfall.

### 4.3. Comparison with Lord Byron's journey

A possible example of use of synthetic weather diary is to track the weather experienced during an expedition or
journey. As an example, I use the famous journey of Lord Byron through the Swiss Alps in September 1816. After
a dreadful summer with almost constant rain, the weather was improving and Lord Byron found the weather to be
quite nice during the trip.
Figure 3 shows the reconstructed fields for five days in September 1816, along with a dot that marks the location
of Byron as well as the weather descriptions form his diary and from the synthetic weather diary, calculated for
each location. The first day („fine weather") indeed was a nice day also in the reconstructions, with no rainfall and
high temperatures. Agreement is also found on the other days, both in terms of rainfall and temperature, except
perhaps for 25 September, when Byron notes "the weather has been tolerable all day" (the meaning of which,
however, is unclear). On this day, the reconstruction shows spatially extensive (though not extremely intense)
rainfall. Also the two stations Aarau and St. Gall report rainfall.

### 5 Discussion

The analyses show that synthetic diaries can provide local, daily weather information in a format that is comparable
to non-instrumental observations and weather descriptions in diaries. The comparison with independent
observations shows some agreement, although the quality both of the daily reconstruction as well as of the
translation needs further improvement.
The comparison of monthly summaries also showed a good agreement and points to the usefulness of vegetation
indices (such as GDD). The monthly summary also points to the effect of combinations of factors (e.g., cold with
little snow) and the importance of pests and insects for agriculture. Experts might be able to make use of the
numerical reconstructions or the synthetic weather data also for analysing insect infestations. In his summaries,
Pictet appears as a rather reserved observer, who largely excludes the societal effects in his descriptions. Other
weather diaries from this time (e.g., the Hoffmann diary, quoted in Bodenmann et al., 2011) have a more
pessimistic or even desperate tone, list in detail the prices, and point to the miserable situation, to beggars etc.
The comparison with the travel diary of Lord Byron yields a general agreement. This shows that synthetic weather
diaries might be useful as an additional information source to better analyse the journey.





For all analyses, we should note that rainfall is much more difficult to reconstruct by the analog method than
temperature due to its very high spatial variability (Pfister et al., 2020). Moreover there is a large representativity
error and arguably also a large observation error. For instance, rain may fall unnoticed during the night. Moreover,
instrumental precipitation (which is the basis for the analog method) is defined as 6 local time to 6 local time,
making comparison at times more difficult; a one-day shift is possible. Note also that the 2x2 km$^2$ grid does not
represent the resolution of the observing network, which has a typical inter-station difference of 15-20 km
(MeteoSwiss, 2019). In any case, precipitation can only be taken as a rough indication. Temperature, conversely,
is well reconstructed, but less often in the focus of observers. Eventually, wind would be an important variable for
any reconstruction, which should be considered in future approaches.
While the agreement as measured in correlation is at times low, currently limiting the application of synthetic
diaries, it should be noted that future reconstructions will likely be much improved and resolve further detail. Once
this stage is reached, the potential of this tool is immense. Weather information can be generated for military
operations or other weather-sensitive activities. Weather diaries can also be produced for travels and expeditions.
The translation into text as well as the comparison with reference periods allows a more direct comparison, and
the calculated indices may be useful for some applications. In particular, impacts on agriculture or on other areas
of life can be assessed more easily. In short, the weather diaries could be used in a variety of ways by historians
to constrain, cross-date, compare or complement other information.
The approach is however also important for sciences. If the translation from numeric weather data to a synthetic
diary that resembles historical sources succeeds, inversion methods can be used to do the opposite: Reconstruct
weather numerically from descriptive data. Data assimilation techniques use such „forward models", however, for
a formal data assimilation approach the variables need to be on a metric scale. For other approaches such as analog
selections, the variables must at least allow to express similarities (or ordinal distances). Systematically compiled
categorial information, however, as is produced in our approach, could be used for instance by machine learning
approaches. This requires that historical weather diaries can be categorized (or are already categorized) in a prior
step. If this is the case, machine learning approaches could then be trained on synthetic weather diaries generated
in the most recent few decades to provide weather types, or even entire weather fields. Once successfully trained,
such approaches can then applied to past weather diaries.
**6 Summary and Conclusions**
Recent efforts in climatology have resulted in daily weather reconstructions for the globe covering the last 200
years. Daily weather reconstructions are also generated for specific regions at high resolution and might soon reach
250 years back. These data sets could be important for history and sciences. In this paper I explore this by
translating a high-resolution, prototype weather reconstruction for Switzerland, 1816, into synthetic weather
diaries for selected locations. These synthetic weather diaries provide not only the reconstructed values, but
translate them into a categorial form that makes them comparable to weather descriptions. Reconstructed values
are referenced to a contemporary reference period, and a calibrated language (e.g., „very cold") is used to translate
numbers to categories. Furthermore, monthly summaries are provided, using the monthly statistics of the daily
reconstruction (translated to descriptive categories) as well as indices for plant growth, freezing, and drought (also
translated to descriptive categories).



Results from our prototype reconstructions for Switzerland, 1816, show a good agreement with independent non-
instrumental daily weather observations from Aarau and St. Gall, and with monthly weather summaries from
Geneva. Also, qualitative agreement with the travel diary of Lord Byron on his journey through Switzerland in
September 1816 is found. The quality of this pilot reconstruction is arguably not accurate enough for many
applications. However, future products are expected to be of sufficient quality to yield useful information, although
there will always be substantial uncertainty for rainfall and particularly for sky conditions.
Synthetic weather diaries are also relevant for science. Combined with machine learning approaches, they could
be used to reconstruct weather numerically from descriptive data. This opens an immense potential for the use of
existing data bases of historical weather data such as EURO-CLIMHIST (Pfister et al., 2017), Tambora.org
(Riemann et al., 2015), and TEMPEST (Veale et al., 2017) but requires intense collaboration between historians
and scientists.

*Acknowledgements:* The work was funded by the Swiss National Science Foundation project WeaR (188701) and
the European Commission through H2020 (ERC Grant PALAEO-RA 787574).

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






Figure 1. Weather observations by Zschokke, Aarau, for July 1816 (Zschokke, 1817).

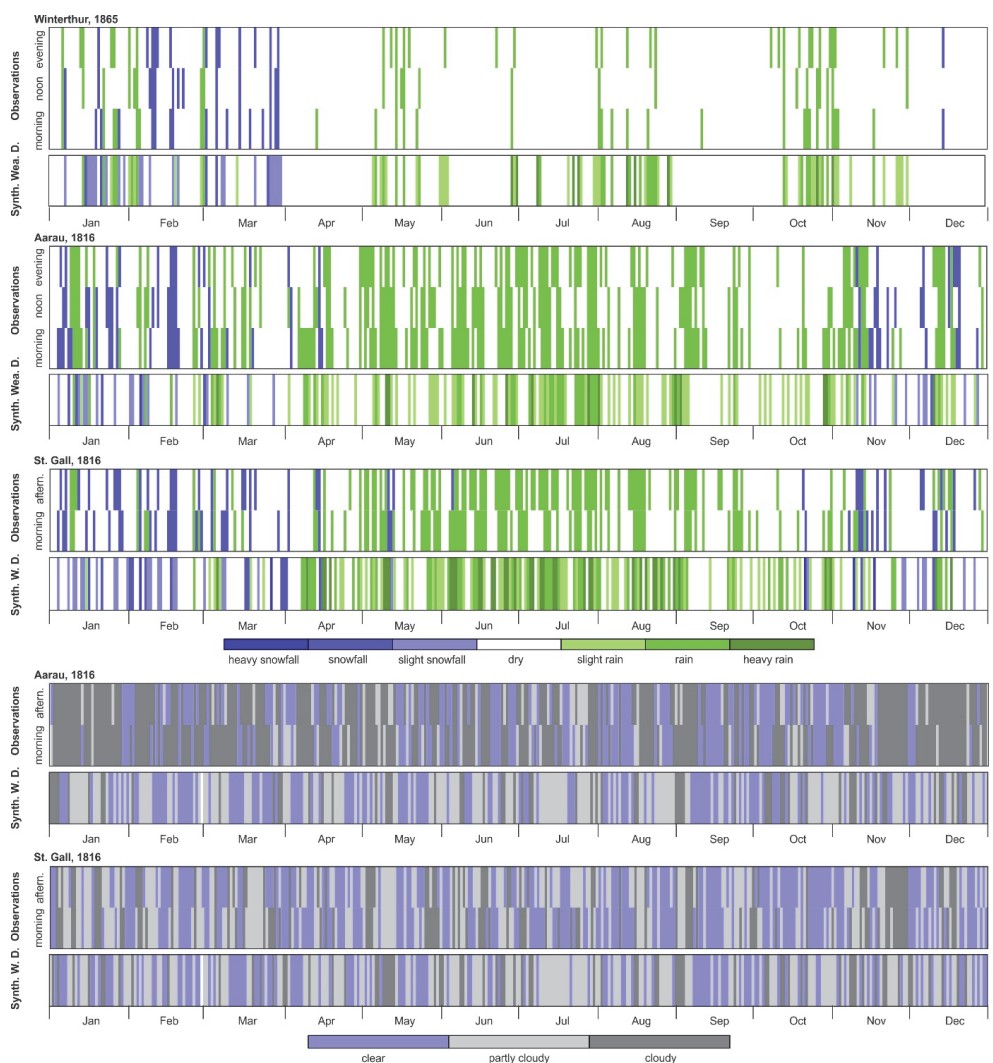

Figure 2. Comparison of observations (2-3 times daily) and synthetic weather diaries (daily) for Winterthur (1865) as well as
Aarau and St. Gall (1816).

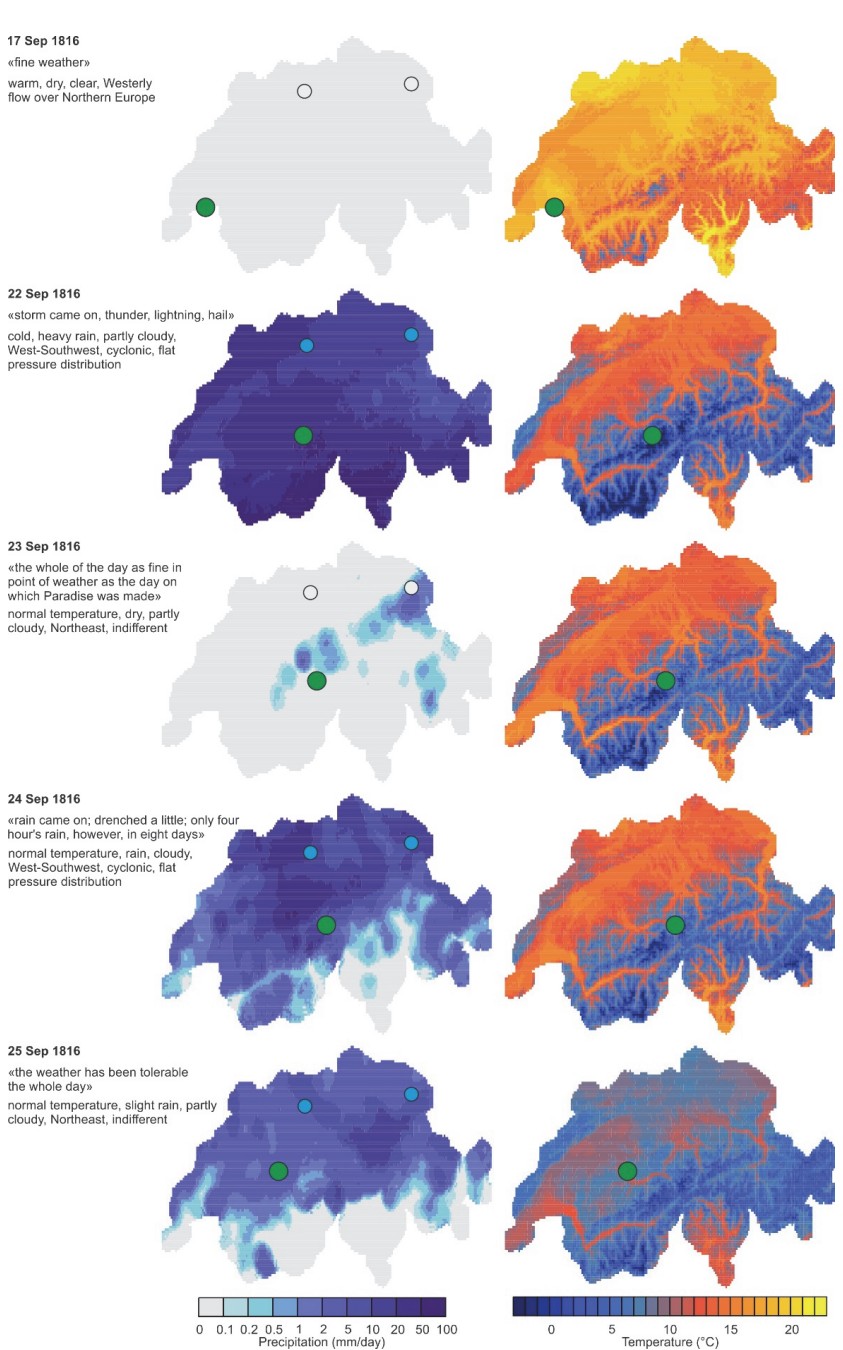

529

Figure 3. The weather during Lord Byron's travel from Lake Geneva to the Bernese Oberland in September 1816.

The figure shows his diary entries (in quotation marks) as well as the synthetic weather diary for five days. Also

shown are precipitation and daily mean temperature from the reconstructions. The green dot marks the position of

Lord Byron. The small dots indicate whether the observers at Aarau and St. Gall noted precipitation (blue) or not

(grey).






Table 1. Description of CAP7 weather types, corresponding CAP9 types, and frequencies in the cold and warm
season in the period 1961-1990

| CAP7 | Description | CAP9 | $f_{Apr-Sep}$ | $f_{Oct-Mar}$ |
|---|---|---|---|---|
| 1 | Northeast, indifferent | 1 | 0.28 | 0.08 |
| 2 | West-southwest, cyclonic, flat pressure | 2 | 0.18 | 0.11 |
| 3 | Westerly flow over Northern Europe | 3 | 0.12 | 0.13 |
| 4 | East, indifferent | 4 | 0.15 | 0.12 |
| 5 | High pressure over Europe | 5+8 | 0.03 | 0.30 |
| 6 | North, cyclonic | 6 | 0.15 | 0.09 |
| 7 | Westerly flow over Southern Europe, cyclonic | 7+9 | 0.09 | 0.16 |


Table 2. Taxonomy used in the observations from Zschokke and Mayer as well as in the synthetic weather diary,
together with the definition (Def.; $T$ = daily mean temperature in °C, $T'$ = standardized daily mean temperature
anomaly, $R$ = precipitation in mm per day, $I$ = Irradiance relative to possible irradiance).

| Temperature | | Precipitation | | | | Sky conditions | | | |
|---|---|---|---|---|---|---|---|---|---|
| Synthetic | Def. | Zschokke | Mayer | Synthetic | Def. | Zschokke | Mayer | Synthetic | Def. |
| extremely cold | $T_{mean}' \leq -2.5$ | (empty), Staub-regen | (empty) | dry | $R<1$ | heiter | schön, heiter, neb. schön | clear | $I<0.33$ |
| very cold | $-2.5<T_{mean}' \leq -1.5$ | Regen | Regen | slight rain | $1<R<5$ $T>2$ | halbheiter | vermischt, neb. vermischt | partly cloudy | $RR>0$ or $0.33<I<0.66$ |
| cold | $-1.5<T_{mean}' \leq -0.5$ | | | rain | $5<R<15$ $T>2$ | bewölkt | trüb, neb. trüb | cloudy | $0.66<I$ |
| average | $-0.5<T_{mean}' \leq -0.5$ | | | heavy rain | $15<R$ $T>2$ | | | | |
| warm | $0.5<T_{mean}' \leq 1.5$ | Schnee, Schnee-regen | Schnee | slight snowfall | $1<R<5$ $T \leq 2$ | | | | |
| very warm | $1.5<T_{mean}' \leq 2.5$ | | | snowfall | $5<R<15$ $T \leq 2$ | | | | |
| extremely warm | $-2.5<T_{mean}'$ | | | heavy snowfall | $15<R$ $T \leq 2$ | | | | |




Table 3. Seven-point scale for indexing standardized anomalies (std. dev.), description (*x* denotes a property and –*x* its reverse,
e.g., "warm" and "cold", "wet" and "dry", or "late" and "early") and corresponding probabilities (Prob.) and cumulative
probabilities (Cum. prob.) for (left part of table) this study and (right) (Pfister et al. 2018). Italics: Thresholds and descriptions
that are part of the definition; other columns are implied by assuming a normal distribution.

| This study | | | | Pfister et al. 2018 | | | |
|---|---|---|---|---|---|---|---|
| Std. Dev. | Description | Prob. (%) | Cum. prob. (%) | Std. Dev. | Description | Prob. (%) | Cum. prob. (%) |
| *< -2.5* | *extremely [-x]* | 0.6 | 0.6 | *< -1.38* | *extremely [-x]* | *8.3* | *8.3* |
| *-2.5 to -1.5* | *very [-x]* | 6.1 | 6.7 | -1.38 to -0.67 | *[-x]* | 16.7 | 25 |
| *-1.5 to -0.5* | *[-x]* | 24.2 | 30.9 | -0.67 to -0.2 | *rather [-x]* | 17 | 42 |
| *-0.5 to 0.5* | *normal/average* | 38.3 | 69.1 | -0.2 to 0.2 | *normal/average* | 16 | 58 |
| *0.5 to 1.5* | *[x]* | 24.2 | 93.3 | 0.2 to 0.67 | *rather [x]* | 17 | 75 |
| *1.5 to 2.5* | *very [x]* | 6.1 | 99.4 | 0.67 to 1.38 | *[x]* | *16.7* | *91.7* |
| *>2.5* | *extremely [x]* | 0.6 | 100 | *>1.38* | *extremely [x]* | *8.3* | *100* |





Table 4. Synthetic weather diary for Aarau, July 1816 (columns: Year, month, day, $T_{mean}$, anomaly of $T_{mean}$, standardized anomaly of $T_{mean}$,
precipitation, temperature description, precipitation description, cloud cover description, weather type, description of weather type.

| YR | M | D | T[°C] | dT[°C] | T[sd] | R[mm/d] | T[text] | R[text] | Sky[text] | WT | WT[text] |
|----|---|---|-------|--------|-------|---------|---------|---------|-----------|-----|----------|
| 1816 | 7 | 1 | 16.40 | -2.25 | -0.69 | 9.69 | cold | rain | partly cloudy | 7 | Westerly flow over Southern Europe, cyclonic |
| 1816 | 7 | 2 | 11.75 | -7.23 | -2.23 | 0.72 | very cold | dry | clear | 6 | North, cyclonic |
| 1816 | 7 | 3 | 12.65 | -6.13 | -1.89 | 0.18 | very cold | dry | cloudy | 6 | North, cyclonic |
| 1816 | 7 | 4 | 14.34 | -4.24 | -1.31 | 0 | cold | dry | clear | 6 | North, cyclonic |
| 1816 | 7 | 5 | 13.31 | -4.52 | -1.40 | 0.72 | cold | dry | clear | 2 | West-Southwest, cyclonic, flat pressure distribution |
| 1816 | 7 | 6 | 13.89 | -4.25 | -1.32 | 10.72 | cold | rain | cloudy | 2 | West-Southwest, cyclonic, flat pressure distribution |
| 1816 | 7 | 7 | 16.92 | -0.50 | -0.15 | 5.43 | normal | rain | cloudy | 7 | Westerly flow over Southern Europe, cyclonic |
| 1816 | 7 | 8 | 18.45 | 0.41 | 0.13 | 19.37 | normal | heavy rain | partly cloudy | 6 | North, cyclonic |
| 1816 | 7 | 9 | 19.55 | 1.39 | 0.43 | 0 | normal | dry | clear | 7 | Westerly flow over Southern Europe, cyclonic |
| 1816 | 7 | 10 | 19.85 | 1.43 | 0.45 | 2.57 | normal | slight rain | partly cloudy | 7 | Westerly flow over Southern Europe, cyclonic |
| 1816 | 7 | 11 | 17.10 | -1.70 | -0.53 | 3.8 | cold | slight rain | partly cloudy | 6 | North, cyclonic |
| 1816 | 7 | 12 | 15.82 | -1.91 | -0.60 | 3.8 | cold | slight rain | partly cloudy | 7 | Westerly flow over Southern Europe, cyclonic |
| 1816 | 7 | 13 | 15.10 | -2.99 | -0.94 | 9.46 | cold | rain | partly cloudy | 6 | North, cyclonic |
| 1816 | 7 | 14 | 11.80 | -5.41 | -1.71 | 2.71 | very cold | slight rain | partly cloudy | 2 | West-Southwest, cyclonic, flat pressure distribution |
| 1816 | 7 | 15 | 15.38 | -2.52 | -0.80 | 10.07 | cold | rain | partly cloudy | 7 | Westerly flow over Southern Europe, cyclonic |
| 1816 | 7 | 16 | 12.14 | -4.94 | -1.57 | 9.69 | very cold | rain | partly cloudy | 7 | Westerly flow over Southern Europe, cyclonic |
| 1816 | 7 | 17 | 10.09 | -6.62 | -2.11 | 3.8 | very cold | slight rain | partly cloudy | 7 | Westerly flow over Southern Europe, cyclonic |
| 1816 | 7 | 18 | 15.18 | -1.44 | -0.46 | 5.89 | normal | rain | partly cloudy | 7 | Westerly flow over Southern Europe, cyclonic |
| 1816 | 7 | 19 | 16.92 | -0.60 | -0.19 | 7.32 | normal | rain | partly cloudy | 2 | West-Southwest, cyclonic, flat pressure distribution |
| 1816 | 7 | 20 | 16.25 | -1.18 | -0.38 | 11.59 | normal | rain | partly cloudy | 2 | West-Southwest, cyclonic, flat pressure distribution |
| 1816 | 7 | 21 | 20.49 | 2.69 | 0.87 | 0.91 | warm | dry | clear | 7 | Westerly flow over Southern Europe, cyclonic |
| 1816 | 7 | 22 | 17.28 | -0.35 | -0.11 | 0 | normal | dry | clear | 6 | North, cyclonic |
| 1816 | 7 | 23 | 15.57 | -2.07 | -0.67 | 0 | cold | dry | clear | 7 | Westerly flow over Southern Europe, cyclonic |
| 1816 | 7 | 24 | 16.35 | -0.98 | -0.32 | 7.06 | normal | rain | cloudy | 7 | Westerly flow over Southern Europe, cyclonic |
| 1816 | 7 | 25 | 13.19 | -4.66 | -1.53 | 9.19 | very cold | rain | partly cloudy | 7 | Westerly flow over Southern Europe, cyclonic |
| 1816 | 7 | 26 | 12.82 | -4.81 | -1.58 | 9.46 | very cold | rain | partly cloudy | 6 | North, cyclonic |
| 1816 | 7 | 27 | 14.31 | -4.77 | -1.57 | 10.19 | very cold | rain | partly cloudy | 1 | Northeast, indifferent |
| 1816 | 7 | 28 | 17.91 | -0.97 | -0.32 | 7.32 | normal | rain | partly cloudy | 2 | West-Southwest, cyclonic, flat pressure distribution |
| 1816 | 7 | 29 | 18.68 | -0.17 | -0.06 | 2.01 | normal | slight rain | partly cloudy | 7 | Westerly flow over Southern Europe, cyclonic |
| 1816 | 7 | 30 | 12.41 | -6.67 | -2.23 | 38.48 | very cold | heavy rain | cloudy | 7 | Westerly flow over Southern Europe, cyclonic |
| 1816 | 7 | 31 | 15.40 | -4.75 | -1.59 | 12.13 | very cold | rain | partly cloudy | 7 | Westerly flow over Southern Europe, cyclonic |





Table 5. Comparison of monthly summaries of weather and state of vegetation by Pictet in Geneva, 1816 (translated), and in our synthetic
weather diary (excerpt giving monthly mean temperature, number of frost days (if any), max₁₀FDD, GDD, delay, monthly mean rainfall,
percentage of normal rainfall, number of rain and snowfall days, P-E with qualifier, overview of frequent (>150%) and infrequent (<50%
relative to expected frequency from reference period) weather types, with number and percentage compared to reference period). Highlighted
in italics are text excerpts (left) that can qualitatively be compared with the synthetic diary (right).

| Mon | Pictet | Synthetic |
|---|---|---|
| Mar | The *harsh temperature*, and the *lack of snow* on the wheats, makes one fear that they will not recover from the winter's hardship. Clovers and alfalfa are uprooted in the cold and wet soils. Field work has been little interrupted. | T = 3.67 °C (*cold*), 3 frost days, max₁₀FDD = 1 °C, GDD = 35 °C (late, 5 d), R = 1.17 mm/d (*56%*), 7 raindays, 3 snowfall days, P-E = 21 mm (normal moisture), frequent „Northeast, indifferent" (8 days, 252%), „Westerly flow over Southern Europe, cyclonic" (9 days, 169%), infrequent „High pressure over Europe" (1 day, 16%) |
| Apr | The season is uniquely *delayed*. The vine has not yet grown at all. The vegetation is very weak. The rains seem to have replenished the meadows that the frosts have lightened; but the wheat is very meager; some have been lost. Spring sowing has been done with ease. | T = 7.86 °C (cold), max₁₀FDD = 1 °C, GDD = 158 °C (*very late,* 21 d), R = 1.77 mm/d (72%), 9 raindays, P-E = 13 mm (normal moisture), frequent „Westerly flow over Southern Europe, cyclonic" (12 days, 202%), infrequent „Westerly flow over Northern Europe" (2 days, 41%) and „West-southwest, cyclonic, flat pressure" (1 day, 36%), no „East, indifferent" and „High pressure over Europe" |
| May | The scourge of cockchafers has been felt with great violence this year. The stone fruit trees are almost completely leafless: plum and cherry trees in particular. Pear trees suffer just as much from caterpillars. The season is *about one month later than last year*. Wheat that has not been destroyed or damaged by the winter has gained a lot. A rather large number of grapes have appeared, but the *unfavourable temperature* cost part of the whites: the red grapes resist better. The meadows look good. | T = 12.29 °C (*cold*), GDD = 415 °C (*very late, 15 d*), R = 2.61 mm/d (104%), 12 raindays, P-E = 6 mm (normal moisture), frequent „North, cyclonic" (16 days, 337%), „Westerly flow over Southern Europe, cyclonic" (7 days, 223%), no „West-southwest, cyclonic, flat pressure", „Westerly flow over Northern Europe", „East, indifferent", and „High pressure over Europe" |
| Jun | Trees are still very susceptible to attacks by cockchafers and caterpillars. The oaks have not yet had a single leaf as of June 30. There are pear trees that also lack them, and whose fruit has fallen off. The wheat has flourished, the barley and oats are beautiful. *The grapes are not yet flowering.* The natural and artificial meadows give a lot of fodder. | T = 12.73 °C (very cold), GDD = 676 °C (*very late, 17 d*), R = 5.51 mm/d (192%), 15 raindays, P-E = 86 mm (very wet), frequent „North, cyclonic" (16 days, 301%), „Westerly flow over Southern Europe, cyclonic" (6 days, 285%), infrequent „Northeast, indifferent" (4 days, 47%), no „Westerly flow over Northern Europe", „East, indifferent", and „High pressure over Europe" |
| Jul | The *cold, rainy weather delayed* the harvest so much that only little rye and little winter barley was harvested. Grapes are *very late* and the branches have a lot of aborted berries, and are also in small quantities. The late-ripening meadows, which are only cut once, yield very little, and the annual clovers look good. Potatoes are in danger of *rotting in places where there is no drainage.* | T = 15.1 °C (*very cold*), GDD = 1020 °C (*extremely late, 22 d*), R = 6.23 mm/d (*254%*), 19 raindays, P-E = 96 mm (*extremely wet*), frequent „Westerly flow over Southern Europe, cyclonic" (16 days, 1385%), „North, cyclonic" (8 days, 162%), infrequent „Northeast, indifferent" (1 day, 9%), no „Westerly flow over Northern Europe", „East, indifferent", and „High pressure over Europe" |
| Aug | The harvests, at first thwarted by the *rains*, were then carried out in very favourable weather. The clovers of the second cut are beautiful, as well as the second-growth hay, and the annual clovers. Potatoes are abundant where they have not *rotted in the ground, which has happened everywhere where water has stayed*. Wheats have little grain and are afflicted with rust. Barley yields a lot. Grapes have grown, but it is doubtful that they can ripen. | T = 16.46 °C (very cold), R = 3.19 mm/d (*119%*), 12 raindays, P-E = 1 mm (wet), frequent „Westerly flow over Southern Europe, cyclonic" (4 days, 329%), infrequent „Westerly flow over Northern Europe" (1 day, 26%), no „High pressure over Europe" |
| Sep | The weather was beautiful and the *temperature was quite mild* throughout the month to advance the ripening of the oats and barley from the mountains: we now have hopes that they will ripen, but the grapes are still almost unchanged, and benefit little. The second-growth hay has been reliable. Sowing is difficult in the clay fields. The first wheat sown | T = 15.04 °C (*average*), R = 2.35 mm/d (78%), 7 raindays, P-E = -5 mm (normal moisture), frequent „Northeast, indifferent" (9 days, 152%), no „High pressure over Europe" and „North, cyclonic" |



| | | |
|---|---|---|
| | has risen well. The potatoes planted by the plough are largely rotten, those planted by the spade are less affected. | |
| Oct | The month of October was of a *remarkable beauty*; the buckwheat sown after the wheat has prospered a lot; the harvest of the mountains is coming, the white grapes got lighter, when the frosts of the 22nd and 23rd spoiled everything, except for the grapes of some vineyards located near the lake. All the red grapes were frozen, because they were just beginning to change. A reliable white harvest was made, and some owners put sugar in it to make sure that the juice would ferment. The wood of the vine is not ripe. | T = 9.23 °C (*cold*), R = 2.59 mm/d (*83%*), 8 raindays, P-E = 41 mm (normal moisture), frequent „Northeast, indifferent" (6 days, 230%), „North, cyclonic" (5 days, 269%) |
