# Peer review of "Synthetic Weather Diaries: Concept and Application to Swiss Weather in 1816"

_Climate of the Past, 2020_

## Referee Comment (RC1) · Anonymous Referee #1 · 7 Jul 2020

To reliably estimate the natural range of changes and variability of climate in the pre-industrial period, early-instrumental meteorological data and weather descriptions are needed, in particular for the last millennium. In recent years a significant growth in this kind of activity (called "data rescue") is observed, including a rising number of publications. Data rescue is important, but equally important are the methods used to conduct weather and climate reconstructions based on gathered information of different categories. The reviewed paper seems very important in improving methods of weather and climate reconstructions using historical data. In the paper the author presents methods to translate numerical reconstructions and derived indices into text describing daily weather and the state of vegetation, which was called a "synthetic weather diary". The analogue method was used for this purpose. All procedure stages leading to con-

struction of the "synthetic weather diary" at daily and monthly resolution are precisely described. In the paper, the application of the proposed method is documented based on available daily weather reconstructions for Switzerland for the year 1816, known as a "Year Without a Summer". Independent man-made documentary data were used to check how good the presented new concept for weather and climate reconstructions is. For temperature comparison it showed quite good agreement, while for precipitation and other study variables it was significantly less so. It seems to me that the proposed method will give significantly better results for areas with less variability in relief than occurs in Switzerland. This is a first, pilot paper that presents some initial results of the proposed new concept for climate reconstruction. The main advantage of this method, which may in future bring revolutionary progress for climate reconstructions, is its possibility to be used to train artificial intelligence (machine learning). As a result, it will be a possibility to reconstruct weather numerically and objectively, based on descriptive data. In the paper, the author gives an extensive and complete, as well as scientifically and methodically correct, analysis of data elaboration and climate reconstructions. The paper is clearly written, well-structured and well documented. Generally I have only a few small suggestions, listed below:

Line 106 – there is: "2.2. Data for comparison: Non-instrumental observations" change to: 2.2. Data for comparison: Man-made documentary evidence For comparison purposes you have used both instrumental measurements and non-instrumental observations available for the Aarau and St. Gall stations. Thus, according to Pfister's nomenclature we should call them "man-made documentary evidence". Line 247 – there is: "4.1 Non-instrumental observations for Aarau and St. Gall" change to: Man-made documentary evidence for Aarau and St. Gall. Lines 382-383 – change ". . .a good agreement with independent non-instrumental daily weather observations from Aarau and St. Gall" to ". . .a good agreement with independent early-instrumental measurements and non-instrumental daily weather observations from Aarau and St. Gall"

In conclusion, I have to say that the paper is very interesting and presents a new concept that may significantly help historians of climate to more efficiently and reliably use the available documentary evidence for reconstruction of climate. I strongly recommend publication of this paper in the journal Climate of the Past. Only minor changes need correcting.

————————————————

---

## Referee Comment (RC2) · Anonymous Referee #2 · 29 Jul 2020

In this very complete and carefully-prepared manuscript, author presented a novel methodology to translate numerical reconstructions into text describing daily weather, termed as "synthetic weather diary". The authors show that "synthetic weather diary" show a good agreement with independent non-instrumental daily weather observation, and monthly weather summaries. This subject is very important for interdisciplinary study between climatologists and historian. Development of global atmospheric re-analysis data back to the 19th century has progressed in recent years. Reviewer think "synthetic weather diary" can be applied as an intermediary between these reanalysis data and regional scale weather studies. Moreover, this methodology has high appli-cability to the other regions of the world like East Asia, where historical daily weather documents are abundant. Reviewer strongly recommend to accept this paper for Cli-

mate of the Past after following minor revision.

Line 118 ,Page4 Autor claims that "St.Gall record contains instrumental(temperature, pressure) and non-instrumental(precipitation, sky cover)information". Usually, we use term of" precipitation" as a observed value by rain gauge in context of climatology. Therefore, it is better to make it clear the definition of the term "non-instrumental" in the present study. Readers not familiar with historical climatology tend to confuse "non-instrumental observation" with "weather dairy".

Figure 2, Author shows comparison between synthetic weather diary and observation. Upper panel of Figure 2 (Winterthur,1865)shows that synthetic weather diary captured more frequent rainfall events in July and August compared to observations. Reviewer would like to recommend to explain possible cause of this difference. In particular, reviewer would like to know whether this difference is typical feature for relatively dry year(1865).

Line 284-287,Page8 Author shows that the difference between relatively dry year(1865) and the wet year(1816) is extremely clear.

If possible, reviewer would like to see the condition for normal year.

---

## Author Comment (AC1) · 5 Aug 2020

Thanks for the review. I will change "non-instrumental observations" to "man-made documentary evidence" at the two instances noted and I will change the sentence (l. 382-383) in the conclusions section.

---

## Author Comment (AC2) · 5 Aug 2020

Stefan Bronnimann

stefan.broennimann@giub.unibe.ch

Thank you very much for the review. I will change the sentence on l. 188 to "twice or three times daily instrumental measurements (pressure, temperature) and non-instrumental observations (whether or not there was precipitation, sky cover)".

Thank you also for your question on the disagreement in July 1865. In fact, from 8 to 31 July, there were no observations, so the plot is misleading and in the revised paper I will mark this period in grey (it is the only missing period in that year). The corrected Figure is attached as Fig. 1. A sentence is also added to the paper.

Line 284-287: I have prepared the figure also for the years 1864 and 1866, which were closer to average in terms of precipitation. The figure is attached to this review as

[Figure]

Figure 2. Results are very similar as for the dry year (except perhaps for the summer of 1866, which appears to be slightly worse). The figure will be added to the paper as a supplementary figure and a sentence of discussion will be added.
* * *
[Figure]

[Figure]

**Fig. 1.** Synthetic diary and observations with respect to precipitation for Winterthur, 1865 (grey = no observations)

[Figure]

**Fig. 2.** Synthetic diary and observations with respect to precipitation for Winterthur, 1864 and 1866 (grey = no observations)

---

## Author Response (AR1)

Dear Editor

Thank you for your decision letter. I have revised the manuscript according to my authors'
replies.

In response to the comments fro mthe Editors, I have added the following sentences to the
discussion section:
"Daily weather reconstructions could also be produced for other regions in Europe with the
analog approach, provided that high-resolution data sets are available for a recent period as a
pool of analogs (for Europe, E-OBS provides daily fields back to 1950 at 0.1° resolution,
Haylock et al., 2008). Daily high-resolution weather reconstructions could also be produced
from dynamical downscaling of reanalyses (Slivinski et al., 2019). Once such high-resolution
daily weather reconstructions are available, the potential is immense."

As to the second comment on climate change, I found it quite difficult to accomodate climate
change in this methodological paper.

Many thanks for handling the manuscript.

Stefan Brönnimann